# Clinically Translatable Approaches of Inhibiting TGF-β to Target Cancer Stem Cells in TNBC

**DOI:** 10.3390/biomedicines9101386

**Published:** 2021-10-04

**Authors:** Andrew Sulaiman, Sarah McGarry, Sai Charan Chilumula, Rohith Kandunuri, Vishak Vinod

**Affiliations:** 1Department of Basic Science, Kansas City University, 1750 Independence Ave, Kansas City, MO 64106, USA; src78797@kansascity.edu (S.C.C.); rohithk@kansascity.edu (R.K.); vishakcollege@gmail.com (V.V.); 2Children’s Mercy Hospital, Kansas City, 2401 Gillham Rd, Kansas City, MO 64108, USA; semcgarry@cmh.edu

**Keywords:** triple-negative breast cancer, cancer stem cell, TGF-β

## Abstract

Triple-negative breast cancer (TNBC) is a subtype of breast cancer that disproportionally accounts for the majority of breast cancer-related deaths due to the lack of specific targets for effective treatments. In this review, we highlight the complexity of the transforming growth factor-beta family (TGF-β) pathway and discuss how the dysregulation of the TGF-β pathway promotes oncogenic attributes in TNBC, which negatively affects patient prognosis. Moreover, we discuss recent findings highlighting TGF-β inhibition as a potent method to target mesenchymal (CD44^+^/CD24^−^) and epithelial (ALDH^high^) cancer stem cell (CSC) populations. CSCs are associated with tumorigenesis, metastasis, relapse, resistance, and diminished patient prognosis; however, due to differential signal pathway enrichment and plasticity, these populations remain difficult to target and persist as a major barrier barring successful therapy. This review highlights the importance of TGF-β as a driver of chemoresistance, radioresistance and reduced patient prognosis in breast cancer and highlights novel treatment strategies which modulate TGF-β, impede cancer progression and reduce the rate of resistance generation via targeting the CSC populations in TNBC and thus reducing tumorigenicity. Potential TGF-β inhibitors targeting based on clinical trials are summarized for further investigation, which may lead to the development of novel therapies to improve TNBC patient prognosis.

## 1. Introduction

A Global Cancer Observatory (GLOBOCAN) study in 2020 demonstrated that there were approximately 19.31 million new cancer cases and 9.96 million cancer-related deaths worldwide [1,2]. In line with these shocking numbers, a recent report by Dagenais et al. demonstrated that while cardiovascular disease is still the number one cause of mortality (40%) worldwide, in high-income countries, deaths attributed to cancer (55%) exceeded deaths due to cardiovascular disease (23%) among adults aged 35–70 [3]. Together, these data suggest that in the developed world, and probably in the future for other nations, cancer will overtake cardiovascular disease as the leading cause of mortality, making the treatment and research of this disease a major medical priority [3].

Further breakdown of the GLOBOCAN 2018 study revealed over 2 million breast cancer diagnoses and over 600,000 breast cancer-related mortalities that year. Thus, breast cancer is the most frequent cancer affecting women, accounting for 1 in 4 cancer cases amongst the female population throughout the world and this disease remains the leading cause of cancer-related deaths amongst women [1]. Triple-negative breast cancer (TNBC) only accounts for 15–20% of breast cancer incidences; however, this subtype is disproportionally associated with decreased patient prognosis and relapse in comparison [4,5]. Additionally, in comparison with other breast cancer subtypes, TNBC due to lack of expression of the estrogen receptor, progesterone receptor, and HER-2 is primarily treated with surgery and non-specific chemotherapy and radiotherapy regimens. As such, the combination of a highly aggressive breast cancer subtype paired with inadequate treatment options contributes towards the dismal prognosis of TNBC compared to other breast cancer subtypes. Treatment for TNBC remains an unmet medical need and the development of novel approaches/therapeutics is required to overcome this hurdle.

### 1.1. Overview of TGF-Β Signaling

In brief, TGF-β signaling is mediated primarily through SMAD or non-SMAD mechanisms [6]. There are three main isoforms of TGF-β (TGF-β1, TGF-β2, and TGF-β3); however, in mammals, TGF-β1 is the predominant isoform and its inactivated form is secreted by cells and bound to extracellular proteins [7]. Various proteins and conditions have been found to activate TGF-β such as pH, ROS, plasminogens, metalloproteinases, and thrombospondin [6,8,9]. Activated TGF-β then binds to the TGF-β type II serine/threonine kinase receptor, which recruits, dimerizes, and phosphorylates the TGF-β type I receptor, promoting its activation. Activated TGF-β type I receptor then phosphorylates and activates SMAD2 and SMAD3. Following their activation, SMAD2 and SMAD3 trimerize with co-SMAD4. The activated SMAD transcription complex then translocates to the nucleus and induces transcription of numerous target genes regulating extracellular matrix production, inflammation, proliferation, immunoregulation and survival (Figure 1) [6,8,9].

TGF-β signaling can also be mediated through non-SMAD-dependent mechanisms through direct phosphorylation of various proteins by the activated TGF-β type 1 receptor. It has been found that TGF-β signaling can then promote MAPK/ERK, PI3K/Akt/mTOR/S6K, RhoA/Rac signaling [10,11,12]. As these pathways are deregulated in TNBC, due to its position as a key branchpoint regulator of these downstream pathways, the modulation of TGF-β may demonstrate to have potent therapeutic effects in the treatment of TNBC [13,14].

### 1.2. The Complicated Regulation of TGF-β in Cancer

TGF-β has been demonstrated to play a biphasic role during tumorigenesis, where it has been demonstrated to act as both a tumor suppressor during early carcinoma development and as a tumor promoter during late carcinoma progression [15]. While outside the scope of this review, Akhurst et al. and Principe et al., amongst others, highlight the complexities, mechanisms and clinical implication of TGF-β signaling and its biphasic regulation of tumorigenesis known as the ‘TGF-β Paradox’ [16,17,18,19,20,21].

During the early stage of tumor development, TGF-β1 has been demonstrated to act as a tumor suppressor through induction of p21CIPI, which prevents cell cycle progression, and through the suppression of c-Myc [15]. TGF-β1 is capable of stimulating apoptosis through SMAD-dependent activation of GADD45b, which then binds and activates p38 triggering programmed cell death [22]. TGF-β1 can also modulate apoptosis through the regulation of pro-apoptotic (BIM and BIK) and anti-apoptotic factors (Bcl-xL) [23,24,25,26]. Tang et al. demonstrated, using xenograft models of early-stage breast cancer, that TGF-β induced differentiation through the downregulation of inhibitor of differentiation/DNA binding (ID1). ID1 is a member of the helix-loop-helix protein family, which binds to basic helix-loop-helix transcription factors to inhibit differentiation and promote self-renewal [27,28]. Using an in vivo serial dilution assay, Tang et al. observed that compared to the control MCF10A-Ca1h xenografts (a xenograft model of human breast cancer using the immortalized human breast epithelial cell line, MCF10A, which has an additional activated Ha-ras on-cogene for tumorigenesis), TGF-βunresponsive tumors, through transfection of a dominant negative type II TGF-β receptor, were 10–20-fold more effective at tumor formation, supporting the tumor suppressor role of TGF-β in early carcinoma development [27].

While TGF-β takes on tumor-suppressive roles during early carcinoma development, it has been found that in various late-stage models of cancer (including breast, prostate, lung, and colorectal cancers), TGF-β signaling is associated with angiogenic, proliferative, and pro-metastatic phenotypes [15,29,30,31,32]. The exact mechanism behind this process remains convoluted; however, it has been found that as cancer progresses, mutations within the TGF-β ligands, receptors and downstream/upstream mediators affecting signaling are widespread and promote dysregulation [33,34,35]. One such example is p53. Upon p53 mutation (one of the most frequently occurring mutations in cancer), TGF-β signaling switched from a tumor suppressor to instead promoting migration and proliferation in ovarian cancer cell line models [33]. A report by Ji et al. sheds light on the complicated crosstalk between p53 and TGF-β, where, using non-small-cell lung carcinoma (H1299) and mouse oral cancer-derived (J4708) cells (both p53-/-), it was demonstrated that transfection of mutant p53 (R175H) binds to the MH2 domain in SMAD3, which led to the disruption of the formation of the SMAD3–4 complex [36]. This correlated with increased migration and proliferation with reduced responsiveness upon TGF-β administration, whereas TGF-β addition to control cells induced the expression of p21WAF1 and suppressed growth and migration [36]. Compared to the controls, gene analysis demonstrated that mutant p53 cell lines decreased the expression of p21 and p15 tumor suppressors upon TGF-β stimulation; however, the gene expression of MMPs and Slug was increased compared to the control, which was correlated with enhanced cellular migration [36]. Treatment with SB431542 (a TGF-β/ALK4/5 inhibitor) restored TGF-βinduced gene expression in both the control and p53 mutant cell lines [34]. Furthermore, siRNA knockdown of SMAD3 demonstrated similar results upon TGF-β stimulation, revealing that it was through p53 antagonism of SMAD3 that TGF-β dysregulation was mediated [36]. Furthermore, mechanistic analysis revealed that it was through ERK signaling that mutant p53 was associating with SMAD3 and, upon inhibition of MEK and ERK, the interaction between mutant p53 and SMAD3, alongside aberrant signaling, was abolished [36]. Together, this research highlights the complicated network facilitating proper TGF-β tumor suppression, how this pathway may be deregulated, the antagonistic role of SMAD3 towards Slug and MMP expression, and how deregulation of this pathway may affect cellular proliferation, migration, and even malignancy.

Other pathways have also been found to modulate TGF-β signaling; it was found that the Akt protein physically interacts with SMAD3, translocating it outside the nucleus and preventing signaling, thus halting TGF-βmediated apoptosis, highlighting that dysregulated P13K/Akt signaling can also alter TGF-β signaling [34]. A recent study by David et al. shed further light on the complicated TGF-β switch in pancreatic ductal adenocarcinoma models [35]. It was demonstrated that TGF-β, through SMAD4, stimulates epithelial to mesenchymal transition (EMT) and migration; however, TGF-β signaling simultaneously promoted apoptosis through upregulation of SNAI1 (an EMT associated factor), which in turn inhibited KLF5, allowing for SOX4 levels to increase and trigger apoptosis [35]. This was interesting, as SOX4 is traditionally associated with tumorigenicity; however, it was found that in a pancreatic ductal adenocarcinoma model, SOX4 induced apoptosis and it was only upon SOX4 complexing with KLF5 (upon downregulation of SNAI1) that there was increased tumorigenesis [35]. This highlights the complicated, contextual balance of TGF-β signaling. As signal modifications are common in cancer, there are a plethora of potential mechanisms that can dysregulate TGF-β signaling, switching it from a tumor suppressor to an oncogene in carcinoma cells. Pro-oncogenic signal pathways such as MAPK, PI3K/Akt/mTOR and c-Myc are also frequently altered in TNBC, which may oppose/antagonize the tumor-suppressive signaling of TGF-β and mechanistically alter the TGF-β pathway [37,38,39]. The studies describing the biphasic role of TGF-β signaling are summarized in Appendix A.

### 1.3. Clinical Correlates of Dysregulated TGF-β Signaling

TGF-β has been found to be negatively correlated with patient prognosis in TNBC. Jiang et al. demonstrated that highly metastatic TNBC is associated with RAB1B (of the RAS oncogene family) suppression. This resulted in elevated TGF-βR1 expression and increased SMAD3 levels and metastasis. When correlated with TNBC patients, it was found that patients with decreased RAB1B expression demonstrated reduced prognosis [40]. 

Ding et al. assessed the correlation between TGF-β signaling and adverse pathological characteristics in TNBC. Amongst the patient samples, 52.5% of TNBC cases were found to express high levels of TGF-β1. Upon assessment, it was found that there was no significant association between TGF-β1 expression and age, menopause, family history or tumor size; however, there was significant association between histological grade (grade III samples; 34 cases in TGF-β1-high samples versus 4 cases in TGF-βlow samples) and positive axillary lymph node tumor migration (33 cases for TGF-β1-high samples versus 16 cases in TGF-βlow samples). Additionally, the 5 year disease-free survival assessment of the patients revealed a substantial decrease in patients with high TGF-β1 expression versus those with low TGF-β1 expression. Moreover, the authors assessed the effects of TGF-β1 exposure using an in vitro TNBC model and it was found that both cellular invasion and metastasis were enhanced once TGF-β1 expression was increased [41]. Thus, patients with increased cytoplasmic TGF-β1 demonstrated a positive correlation with increased tumor grade, lymph infiltration, and diminished disease-free survival, making TGF-β1 a clinically translatable target, which may play a role in patient outcomes [41,42,43]. 

Using cBioportal and the The Cancer Genome Atlas’ (TCGA) PanCancer Atlas in our own analysis, we assessed 1082 breast cancer patients and grouped them into two categories based on TGF-β pathway gene expression (TGF-β high vs. low) [44,45,46,47]. We found that high TGF-β signaling was associated with diminished overall survival (Figure 2, 16.8% mortality with a 122.83 median month survival in TGF-β high vs. 12.7% with a 140.28 median month survival in TGF-βlow groups, * *p* < 0.05). This database analysis supports other studies which demonstrate that TNBC is associated with increased TGF-β signaling. We then stratified the 1082 breast cancer patients into TNBC, HER-2, Luminal A and Luminal B subtypes and found that *TGF-Β1* and *TGF-Β2* mRNA expression was significantly elevated in TNBC patient samples compared to the other subtypes (Figure 2B). Finally, we looked specifically at TNBC breast cancer patients and stratified the population based on low (<0 fold), normal (0–2 fold) or high (>2 fold) *TGF-ΒR1* mRNA expression and found reduced disease-specific survival in TNBC patients with elevated *TGF-ΒR1* mRNA expression (Figure 2C). Together, these data demonstrated that TGF-β signaling is correlated with a reduced patient prognosis, is elevated in TNBC compared to other breast cancer subtypes and is correlated with reduced patient prognosis in TNBC patients, supporting the need for the advancement of therapeutic modulation of TGF-β [41,42,48].

### 1.4. Clinical Importance of CSCs in TNBC

Breast cancer stem cells (CSCs) represent a small percentage of cells within tumors that exhibit stem cell-like properties, such as self-renewal, differentiation, and quiescence [49]. CSCs are at the apex of the cellular hierarchy within tumors, capable of maintaining CSC pools and giving rise to non-CSC bulk tumor cells to promote disease progression, resistance generation, and facilitate tumor metastasis [50,51,52]. 

In breast cancer, there are two major CSC populations which are characterized by CD44^+^/CD24^−^ and ALDH^high^ markers [53,54]. Al Hajj et al. fractionated breast cancer cells using flow cytometry and then through serial dilution assays demonstrated that the CD44^+^/CD24^−^ CSC population showed an impressive 100-fold increased tumorigenicity compared to unfractionated cells [55]. The CD44^+^/CD24^−^ CSC population in breast cancer is associated with a mesenchymal phenotype, increased N-cadherin expression, decreased E-cadherin, and increased YAP, Twist, Snail, and Slug gene expression [53,56,57,58]. This population also demonstrates increased migration, resistance to conventional chemotherapeutics, increased reliance on glycolysis and quiescence [53,56]. 

The ALDH^high^ CSC population is characterized by being able to form a tumor with as little as 1500 breast cancer cells [59,60]. In contrast to the mesenchymal CD44^+^/CD24^−^, ALDH^high^ CSCs demonstrate an epithelial phenotype with high E-cadherin expression, low N-cadherin, vimentin, Slug, Wnt, Twist, and Snail expression [53,57,61]. ALDH^high^ CSCs were found to be highly enriched for HIF-1α signaling, angiogenic promotion and were highly proliferative [53]. Importantly, both epithelial and mesenchymal CSCs possess differential signaling enrichment/repression, can interconvert, exist on a gradient and work together to facilitate metastasis and secondary tumor formation [53,57,62].

Conventional therapy using anthocyanins, taxols, and other antimetabolite or antineoplastic agents, while effective against the bulk population, are ineffective at targeting CSCs and even lead to the enrichment of CSCs post-therapy [57,63,64,65]. This is highlighted by Creighton et al. who demonstrated that in post-chemotherapy breast cancer patients there was an increased frequency of CD44^+^/CD24^−^ CSCs populations compared to the proportion present before treatment [66]. In breast cancer tissue samples post-letrozole treatment it was found that there was an increase in FN1, SNAI2, VIM, FOXC2, MMP2, and MMP3 (mesenchymal-related genes) as well as diminished CDH1 (an epithelial-related gene) suggesting an enrichment of mesenchymal properties and EMT (epithelial to mesenchymal transition) [57,62,66,67,68,69,70]. EMT is a process through which epithelial cells gain mesenchymal properties which correlate into enhanced migration and invasion properties allowing for increased metastasis in cancer models [57,62,66,67,68,69,70]. Creighton et al. provided clinical evidence that post-chemotherapy, CSCs can be enriched and gain a mesenchymal phenotype in breast cancer models [66]. Thus, methods to increase therapeutic efficacy of chemotherapy, to prevent CSC enrichment, to assesses CSC populations before and following treatment may present a useful clinical indicator of therapeutic efficacy. 

Similarly, our own research has been demonstrated in TNBC in vivo mouse models using patient-derived xenografts (patient tumors implanted immediately and only as solid tumors into immunocompromised mice) that post-chemotherapy exposure led to increased CD44^+^/CD24^−^ and ALDH^high^ CSC populations [70]. Afterwards, using a serial dilution assay (the gold standard for functional tumorigenicity), it was found that compared to the control, chemotherapy-treated PDX tumors demonstrated enhanced tumor formative capabilities (forming tumors at a rate of 80% upon an injection of 1,000,000 cells versus the control, which formed tumors at a rate of 20% with an injection of 1,000,000 cells) [70]. These studies demonstrate that chemotherapy induced CSC enrichment represents a major factor in relapse and tumor reconstitution. As such, methods to assess CSC enrichment pre- and post-chemotherapy may be a useful indicator to gauge chemotherapeutic efficacy and assess potential relapse rate and patient prognosis.

Yu et al. illustrated a method to assess these populations using a dual-colorimetric RNA in situ hybridization approach to assess cells for epithelial/mesenchymal gene expression that breast CSCs revealed epithelial, mesenchymal, and epithelial/mesenchymal hybrid signatures [71]. Pre- and post-chemotherapy analysis was performed (post-treatment with cisplatin, taxol, and adriamycin) on circulating tumor population numbers and CSC plasticity [71]. It was found that chemotherapy-responsive patients demonstrated decreased CSCs and a proportional decrease in mesenchymal CSCs in comparison to epithelial CSC populations. In patients with progressive disease, there were increased mesenchymal CSCs and increased multicellular CSC clusters which were also highly positive for mesenchymal markers, thus demonstrating how non-specific chemotherapy can influence CSC plasticity and promote increased tumor cell dissemination [71].

Another report by Papadaki et al. used ALDH1 (an epithelial marker) and Twist (a mesenchymal marker) to determine epithelial, mesenchymal, or epithelial/mesenchymal populations in the CSCs of 130 breast cancer patients [72]. It was found that hybrid epithelial/mesenchymal CSCs were associated with increased rates of lung metastasis, increased rates of patient relapse, and decreased progression-free survival (10.2 months vs. 13.5 months) [72]. Chemotherapy treatment increased hybrid epithelial/mesenchymal CSCs whereas the epithelial and mesenchymal CSCs was reduced [72]. These findings in combination with other reports advocate that chemotherapy treatment alters the plasticity and population dynamics of epithelial, mesenchymal, and epithelial/mesenchymal CSCs, decreases patient prognosis and increases the rates of metastasis/relapse [53,54,57,63,73].

Such findings highlight the magnitude of CSCs in patient outcome, the need for novel therapeutic treatment, and support further studies in investigating CSC enrichment as indicators for patient prognosis. The studies describing the clinical importance of CSCs in TNBC are summarized in Appendix A. 

### 1.5. TGF-β as a Therapeutic Target to Inhibit TNBC and Its CSC Population

TGF-β has been demonstrated to be enriched alongside ALDH^high^ and CD44^+^/CD24^−^ (epithelial, and mesenchymal CSC markers) in chemotherapy-treated TNBC patients [74]. Upon direct administration of paclitaxel to TNBC cell lines, similar results were observed with an increase in tumorigenesis and mammosphere formation [74]. Importantly, it was found that the CSC-enriching effects of paclitaxel chemotherapy were promoted through TGF-β-mediated SMAD4-dependent expression of IL-8. Upon siRNA inhibition of SMAD4 or exposure to LY2157299 (a TGF-β type I receptor kinase inhibitor), tumorigenesis was rescued and epithelial, and mesenchymal CSC populations were inhibited. These findings were verified in vivo using mouse TNBC tumor models and it was found using serial dilution tumorigenesis assays that compared to the control (3/5 tumors formed at an injection concentration of 1 × 10^3^ cells) paclitaxel treatment increased tumorigenesis (4/5 tumors formed at an injection concentration of 1 × 10^3^ cells), while the combination of paclitaxel and LY2157299 was able to reduce tumorigenicity (2/5 tumors formed at an injection concentration of 1 × 10^3^ cells) [74].

These results correlate with recent findings from Yadav et al., where it was demonstrated in breast cancer cell lines that after treatment with radiotherapy, the surviving cells demonstrated increased rates of proliferation and TGF-β1, TGF-β2 and TGF-β3 expression. Interestingly, these cells also demonstrated increased CSC markers (CD44^+^/CD24^−^/ALDH^high^) and enhanced migration. Further treatment was met with resistance; however, treatment with TGF-β1 inhibitors was able to rescue and re-sensitize cells to radiotherapy [75]. 

Epirubicin is another widely used anthracycline to treat TNBC. It has been shown to cause enriched CD44^+^/CD24^−^ CSCs and tumorigenicity of breast cancer following treatment [76]. A study by Xu et al. transformed MDA-MB-231 TNBC cells (epirubicin-sensitive) into an epirubicin-resistant cell line (MB-231/Epi) through chronic exposure to epirubicin. Resistance was correlated with higher levels TGF-β expression, chemotherapy resistance and CD44^+^/CD24^−^ CSC enrichment. In addition to this, MB-231/Epi cells showed increased migration and invasion which indicated potentially enhanced metastatic potential. Thus, this paper highlights the potential association between TGF-β, chemoresistance and CSC enrichment leading to enhanced tumor progression and metastasis, highlighting the importance of targeting TGF-β in TNBC [77]. 

In concordance with other reports, a study by Zhu et al. found that TGF- β1 treatment in TNBC cells led to increased expression of the mesenchymal markers Vimentin and N-Cadherin, and the decreased the expression of the epithelial marker E-cadherin [78]. This pattern of expression is consistent with the EMT model of metastasis and indicates increased migration, invasion and metastatic potential [53,57]. Additionally, TGF-β1 treatment in TNBC models demonstrated increased resistance to anoikis and increased matrigel invasion in vitro. Mechanistic analysis revealed that TGF-β1-induced cell metastasis via ITGB1 upregulation and downstream FAK autophosphorylation alongside Src activation. Moreover, this FAK/Src signaling led to Akt phosphorylation and eventual β-catenin signaling [78]. Upon ophiopogonin D treatment (an anti-inflammatory agent with TGF-β1 inhibitory properties) TGF-β1-mediated effects on invasion, resistance and metastasis in TNBC models were abrogated through disruption of TGF- β1 stimulation of the ITGB1/FAK/Src/AKT/β-catenin signaling pathway [78]. Treatment with ophiopogonin D LAO led to reduction in TNBC viability and prevention of EMT marker enrichment post-TGF-β1 exposure suggesting reduced metastatic potential. This study identifies both a potential mechanism through which TGF-β signaling promotes metastasis, proliferation and EMT in TNBC models and highlights TGF-β inhibitors as a potent method to alleviate these changes [78].

A study by Sun et al. further looked into the associated between TGF-β, CSC enrichment and radioresistance. Sun et al. demonstrated that following initial radiotherapy, breast cancer patients who demonstrated radioresistance and recurrence within 5 years of their initial therapy were found to have increased expression of alpha-1,3-mannosyltransferase (ALG3) [79]. These findings were correlated with breast cancer cell lines where basal-like and HER-2+ breast cancer lines demonstrated increased levels of radioresistance and ALG3 expression. Moreover, upon the creation of an ALG3-overexpression model, previously radiosensitive breast cancer cell lines demonstrated radioresistance, and ALG3-overexpressing breast cancer cell lines, when injected subcutaneously into mice, displayed an increased tumor growth rate and OCT4 gene expression (a commonly used marker to assess CSC enrichment). Conversely, it was also demonstrated in the basal-like TNBC cell lines that upon ALG3 knockout models, previously radioresistance cell lines were sensitized, tumor growth in vivo was delayed and OCT4 expression was decreased. Further assessment of ALG3 modulation of CSCs in breast cancer demonstrated that ALG3-overexpressing cell lines also demonstrated increased NANOG, OCT4, and SOX2 expression (CSC associated genes) and increased tumorsphere formation capabilities. FACs analysis demonstrated increased CD44^+^/CD24^−^ CSCs in wild-type ALG3-overexpressing breast cancer cell lines; however, this population was severely diminished upon ALG3 knockdown (control MDA MB-231 TNBC cells were 75.3% CD44^+^/CD24^−^, while ALG3 knockdown MDA MB-231 cells were only 42.1% CD44^+^/CD24^−^), highlighting that ALG3 may serve as a potential target to decrease radioresistance in breast cancer [79]. Mechanistic analysis through luciferase assay determined that ALG3 downregulation reduced the luciferase signal of SMAD-luc, demonstrating TGF-β signal modulation via ALG3. Further assessment demonstrated that ALG3 expression promoted the glycosylation of TGF-βR2, which mediated TGF-β signaling. It has previously been demonstrated that glycosylation of TGF-βR2 affects its ligand-binding sensitivity and reduced glycosylation of TGF-βR2 leads to disrupted binding capacity with TGF-βR1, which in turn reduced phosphorylation of SMAD2 and ultimately TGF-β signaling [79,80].

Usage of tunicamycin (a N-linked glycosylation inhibitor) demonstrated similar effects on TGF-βR2 as the ALG3 knockdown cell lines. Finally, co-immunoprecipitation demonstrated an interaction between TGF-βR1 and TGF-βR2, as well as TGF-βR1 and P-smad2 in ALG3-expressing breast cancer cell lines. This co-immunoprecipitation was not observed in ALG3 knockout cell lines. A TGF-βR2 inhibitor (LY2109761) was then used to inhibit ALG2 overexpressing breast cancer cell lines which induced apoptosis post-radiotherapy and diminished tumorsphere formation as well as CD44^+^/CD24^−^ CSCs [79]. 

As indicated through the above studies, CSC enrichment and resistance post-chemotherapy and radiotherapy may be targeted through TGF-β inhibition. Thus, TGF-β signaling may provide a promising target for CSC inhibition in TNBC to be used in conjunction with conventional therapy. Other studies have produced similar findings using TGF-β inhibitors on breast cancer models in vitro and in vivo. Schech et al. demonstrated the efficacy of entinostat (a class I HDAC inhibitor with TGF-β modulating properties) at inhibiting CD44^+^/CD24^−^ CSCs in TNBC cell lines (from 63.1% to 3.66% in MDA MB-231 cells) [81,82]. Additionally, immortalized non-cancerous breast cancer lines (MCF-10a and 184B5) cells were induced to form mammospheres and enrich their CSC population through TGF-β exposure. This effect was inhibited upon treatment with entinostat or LY2109761. Moreover, TNBC cells were inoculated into the fat pads of mice and lung metastasis was assessed after 3 weeks. Mice treated with entinostat demonstrated reduced tumor growth in vivo as well as reduced rates of lung metastasis. 

Another study by Wahdan-Alaswad et al. found that TNBC lines possessed high levels of TGF-β receptors compared to other breast cancer subtypes. Moreover, exposure of TNBC cells to TGF-β1 increased promoted proliferation and increased the expression of phospho-Smad2 (P-Smad2), phospho-Smad3 (P-Smad3) and ID1 protein expression in response [83]. LY2197299 (a selective TGF-β receptor I-kinase inhibitor) was then used to inhibit TGF-β1 signaling alongside metformin (an AMPK activator frequently prescribed for the treatment of type II diabetes mellitus). Predicably, LY2197299 suppressed proliferation in TNBC cells and TGF-β1 signaling. Interestingly, metformin was also capable of suppressing proliferation in TNBC cells at concentrations of 2.5 mM and synergized with LY2197299 in this regard [83]. Moreover, both LY2197299 and metformin were capable of inhibiting phospho-Smad2 and phospho-Smad3 protein expression following treatment [83]. It was found that both metformin and LY2197299 were capable of inhibiting TGF-β1-induced motility and cell invasion in TNBC models. This study demonstrates the importance of assessing commonly used, well-tolerated therapeutics at clinically relevant dosages for TGF-β inhibitory properties [83]. Such a discovery could generate a safe, well-tolerated enhancement to conventional therapy which can lead to increased treatment efficacy and reduced rates of metastasis, resistance and patient relapse.

For future investigations, active interventional clinical trials listed in Clinicaltrials.gov (accessed on 9 September 2021) database for the treatment of patients with various cancers through TGF-β inhibition are summarized in Table 1. These potential TGF-β modulators/inhibitors seem to be safe for usage in the clinic and have been demonstrated to suppress the TGF-β signaling pathway in preclinical studies though their efficacy in the treatment for TNBC remains to be determined. We have also listed completed clinical trials for the treatment of breast cancer with TGF-β inhibitors for further investigation (Table 2). Future translational research to determine the clinical efficacy of TGF-β inhibitors in targeting TNBC CSCs and impeding tumorigenicity in combination with other inhibitors and chemotherapeutic drugs may lead to the development of a tangible therapy to improve patient prognosis. The studies describing the preclinical TGF-β Inhibitors referenced in this section are summarized in Appendix A.

Additionally, with the promising results of immunotherapy in recent years, novel targeting combining immunotherapy and TGF-β inhibition may lead to new avenues of treatment. To our knowledge, currently, there is no combinational treatment consisting of TGF-β modulation with immunotherapy in TNBC models according to clinicaltrials.gov (accessed on 9 September 2021); however, there are trials consisting of galunisertib (LY2157299 monohydrate), a specific small-molecule inhibitor of TGF-βR1 kinase in combination with checkpoint inhibitors in NSCLC, HCC and pancreatic cancer patients (NCT02423343; NCT02734160) [84]. Additionally, this combination has demonstrated preclinical efficacy in 4TI-LP breast cancer cells in vivo (4TI-LP being 4TI cells with luciferase expression) [84]. As such, combinational inhibition of checkpoint inhibitors and TGF-β modulation may be a novel approach for the treatment of breast cancer. This may be a promising approach for future investigation [85,86,87].

**Table 2 biomedicines-09-01386-t002:** TGF-β Inhibitors in Completed Breast Cancer Clinical Trials. The Clinicaltrials.gov (accessed on 9 September 2021) database was used to assess completed, interventional clinical trials for breast cancer/neoplasm treatment within all phases of development. Clinical Trial Search link (accessed on 1 August 2021): https://clinicaltrials.gov/ct2/results?cond=Breast+Cancer&term=tgf&type=Intr&rslt=&recrs=e&age_v=&gndr=&intr=&ttles=&outc=&spons=&lead=&id=&cntry=&state=&city=&dist=&locn=&rsub=&strd_s=&strd_e=&prcd_s=&prcd_e=&sfpd_s=&sfpd_e=&rfpd_s=&rfpd_e=&lupd_s=&lupd_e=&sort=.

Inhibitor	Clinical Trial Number	Mechanism of Action	Tumor Type	Results
Imiquimod [71]	NCT00821964	Inducer of interferon-gamma, known to inhibit TGF-β	Male breast cancer, recurrent breast cancer, skin metastases, stage IV breast cancer	Treatment using TLR-7 agonist, imiquimod and systemic albumin bound paclitaxel for recurrent chest wall lesion in breast cancer is effective in inducing disease regressing with a response rate of 20–30% [85].
Fenretinide	NCT00001378	Retinoid inducing dimerization of retinoid acid receptors which has been shown to regulate affect multiple signal transduction pathways, including IGF, TGF-β, and AP-1 [72]	Breast cancer and neoplasms	No results posted.
Fresolimumab [52]	NCT01401062	A human anti-(TGF-β) monoclonal antibody	Metastatic breast cancer	TGF-β blockade using fresolimumab during radiotherapy in metastatic breast cancer was well tolerated. Patients receiving a higher dosage of fresolimumab (10 mg/kg) had a longer median overall survival compared to lower dosage (1 mg/kg). The higher-dosage arm also expressed greater systemic immune response with improved peripheral mononuclear cell count and increased CD8 central memory pool [88].
Bevacizumab	NCT01959490	Humanized monoclonal antibody that targets VEGF-A and has demonstrated inhibited TGF-β following treatment [73]	Breast cancer stages II–IIIC	60% of patients treated with bevacizumab and combination chemotherapy (doxorubicin + cyclophosphamide + paclitaxel) achieved pathological complete response (pCR, absence of invasive cancer in breast or lymph nodes after neoadjuvant chemotherapy). Data accessed from clinicaltrials.gov (accessed on 9 September 2021).
Galunisertib (LY2157299)	NCT02423343 NCT02178358 NCT02734160 NCT01246986	A small-molecule inhibitor of the TGF-β receptor I kinase	Solid tumors, NSCLC, HCC recurrent	NCT02423343—galunisertib + nivolumab had a progression-free survival of 5.26 and 5.39 months, with an overall survival of 11.99 and 14.52 months in two cohorts. Data accessed from clinicaltrials.gov (accessed on 9 September 2021). NCT02734160—Co-administration of galunisertib with durvalumab resulted in a 25% disease control rate, with a medial overall survival and progression-free survival of 5.72 months (95% CI: 4.01 to 8.38) and 1.87 months (95% CI: 1.58 to 3.09) [89]. NCT01246986—The overall survival of 160 mg treated vs. 300 mg using galunisertib was 7.3 and 16.8 months, respectively. Median overall survival of TGF-β1 responders vs. non-responders was 11.2 and 5.3 months, respectively [90].
Vactosertib (TEW-7197)	NCT02160106	Potent TGF-Β receptor ALK4/ALK5 inhibitor	Advanced-stage solid tumors	No results posted.
NIS793	NCT02947165	mAb that binds to human TGF-Β and prevents of activation of downstream signaling	Breast, lung, hepatocellula, colorectal, pancreatic and renal cancer	No results posted.

### 1.6. Conclusions

For the development of effective therapeutic approaches, future preclinical research must consider targeting both epithelial and mesenchymal CSCs and assess how experimental treatments affect these populations using clinically translatable models. While tumor shrinkage models demonstrate time point efficacy of therapy, CSC composition assessment must be performed to determine whether the investigated therapy reduces or enriches CSC populations within the tumor to determine long-term clinical efficacy. To that end, we advocate for serial dilution assessments and FACS assessment post-therapy to determine tumor population assessment and functional tumorigenicity post-therapy. Furthermore, we endorse multiple rounds of serial dilutions/treatment and CSC assessment may be performed to mimic long-term survival and effects on tumorigenicity with multiple rounds of therapy, which would provide substantial evidence into long-term clinical efficacy and patient prognosis. 

There are currently no specific treatment therapy options for TNBC patients. Given the preclinical and clinical evidence of TGF-β inhibitors, future studies using known and novel regulators of the TGF-β pathway may lead to a clinically translatable breakthrough therapy.

## 2. Materials and Methods

Breast cancer datasets from The Cancer Genome Atlas’ PanCancer Atlas (TCGA, https://www.cell.com/pb-assets/consortium/pancanceratlas/pancani3/index.html, accessed on 29 September 2021) [41] were used and analyzed with cBioportal (http://www.cbioportal.org/index.do, accessed on 29 September 2021). High TGF-β gene expression was defined based on the following gene set available at cbioportal consisting of 30 genes associated with the TGF-β superfamily with the following genes each having an mRNA expression greater than 3 standard deviations above the mean: *TGF-Β1*, *TGF-Β2*, *TGF-Β3*, *TGF-ΒR1*, *TGF-ΒR2*, *TGF-ΒR3*, *BMP2*, *BMP3*, *BMP4*, *BMP5*, *BMP6*, *BMP10*, *BMP15*, *BMPR2*, *ACVR1*, *ACVR1B*, *ACVR1C*, *ACVR2A*, *ACVR2B*, *ACVRL1*, *SMAD2*, *SMAD3*, *SMAD1*, *SMAD5*, *SMAD4*, *SMAD9*, *SMAD6*, *SMAD7*, *BMPR1A*, and *BMPR1B*. Expression data, correlation data, mutational frequency, breast cancer subtype analysis and Kaplan–Meier survival curves were generated using the datasets compiled by June 2020 from the following database IDs: https://bit.ly/2MVN0KN.

Subtypes were stratified based on the subtype category in the same study (BRCA_Basal, BRCA_HER2, BRCA_LumA and BRCA_LumB from https://www.cbioportal.org/study/summary?id=brca_tcga_pan_can_atlas_2018, accessed by 29 September 2021) and *TGF-Β1* and *TGF-Β2* mRNA expression was compared using mRNA Expression, normalized from illumine HiSeq RNASeqV2 (log2). Invasive breast cancer samples were then stratified into TNBC via the BRCA_Basal subtype and *TGF-ΒR1* gene expression (mRNA expression z-scores relative to normal samples (log RNA Seq V2 RSEM) was used at <0, 0–2 and >2 fold to compare disease-free survival.

## Figures and Tables

**Figure 1 biomedicines-09-01386-f001:**
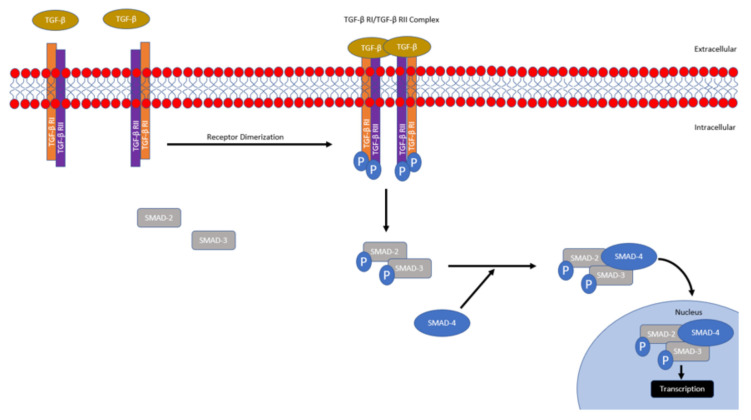
Overview of Conventional SMAD-mediated TGF-β Signaling. Activated TGF-β binds and promotes the dimerization of TGF-β type 1 and type 2 receptors, which leads to the phosphorylation and subsequent activation of the TGF-β type 1 receptor. Activated TGF-β type 1 receptor then phosphorylates SMAD 2 and 3, promoting complex formation with co-SMAD (SMAD4). The SMAD trimer complex then translocates into the nucleus, where it promotes transcription.

**Figure 2 biomedicines-09-01386-f002:**
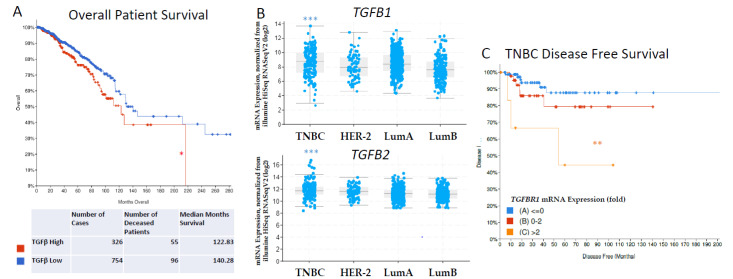
Database Analysis of TGF-β Gene Expression and Survival in Breast Cancer Patients. (**A**) Kaplan–Meier curves for overall survival of the patients with high expression of TGF-β signaling in cancer samples (red curve) in comparison with patients with unaltered expression (TGF-β low, blue curve). *n* = 1082, * *p* = 0.0303, log-rank test. (**B**) Patient breast cancer samples (*n* = 1082) were stratified based on TNBC, HER-2 +, Luminal A and Luminal B subtypes and *TGF-Β1* (*** *p* = 5.42 × 10^−12^) and *TGF-Β2* (*** *p* = 4.19 × 10^−10^) gene expression was assessed. (**C**) Invasive breast cancer samples which were of the TNBC subtype (*n* = 137) were then stratified and separated into categories based on *TGF-ΒR1* mRNA expression (log RNA Seq V2 RSEM) to generate a Kaplan–Meier curves for disease-free survival ** *p* = 8.710 × 10^−3^.

**Table 1 biomedicines-09-01386-t001:** TGF-β Inhibitors in Active Cancer Clinical Trials. The Clinicaltrials.gov (accessed on 9 September 2021) database was used to assess active, interventional clinical trials for cancer treatment within all phases of development. Clinical Trial Search link (accessed on 1 August 2021): https://clinicaltrials.gov/ct2/results?term=tgf&cond=Cancer&flds=abky&Search=Aply&recrs=f&recrs=d&age_v=&gndr=&type=&rslt=.

Inhibitor	Clinical Trial Number	Mechanism of Action	Tumor Type
AVID200	NCT03834662	A receptor ectodomain trap that inhibits TGF-β1 and TGF-β3	Malignant solid tumors
Bintrafusp alfa (M7824)	NCT04246489 NCT04551950 NCT03833661	Bifunctional fusion protein with a ectodomain of TGF-ΒRII fused to human IgG1 blocking PD-L1	Uterine cervical neoplasms, biliary tract cancer, cholangiocarcinoma, gallbladder cancer
Fresolimumab	NCT02581787	A recombinant anti-TGB growth factor antibody against TGF-Β1,2,3	Stage IA-B NSCLC
Vactosertib (TEW-7197)	NCT03732274 NCT04103645	Potent TGF-Β receptor ALK4/ALK5 inhibitor	Metastatic NSCLC Myeloproliferative neoplasm
GT90001	NCT03893695	Fully human anti-ALK1 monoclonal antibody that inhibits TGF-Β and ALK-1	Metastatic HCC
Nimotuzumab	NCT00957086	Recombinant humanized murine immune antibody that blocks TGF and EGF	Squamous-cell carcinoma of the head and neck
LY3200882	NCT02937272	An ATP competitive inhibitor of the serine-threonine kinase domain of TGF-βRI	Solid tumors
Galunisertib (LY2157299)	NCT02423343 NCT02906397 NCT02178358 NCT03206177	A small-molecule inhibitor of the TGF-β receptor I kinase	Solid tumors, NSCLC, HCC recurrent, lung neoplasms, esophageal neoplasms, stomach neoplasms, hepatocellular carcinoma
Losartan	NCT01821729	Angiotensin II receptor antagonist	Pancreatic cancer
M7824	NCT03833661	A fusion protein inhibitor comprised of human TGF-βRII fused to the extracellular domain of human linked to the C-terminus of human anti-PD-L1 heavy chain	Biliary tract cancer, cholangiocarcinoma, gallbladder cancer

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
