# Peer review of "Clinically Translatable Approaches of Inhibiting TGF-β to Target Cancer Stem Cells in TNBC"

_biomedicines, 2021, doi:10.3390/biomedicines9101386_

Round 1

Reviewer 1 Report

The review by Andrew Sulaiman et al. describes the impact of the TGFb pathway in the behavior of triple-negative breast cancer (TNBC). Specifically, it shows its impact at early and late stages and how this pathway regulates the different subtypes of cancer stem cells (CSC) described in TNBC. Moreover, describes how current chemotherapy and radiotherapy treatments impact this pathway and therefore, in the aggressiveness of CSC and the outcome of patients. The review is well-written and easy to read. It is interesting from a clinical point of view as it suggests that novel studies inhibiting this pathway might be beneficial for the treatment of TNBC patients.

Some changes might be done to improve the quality of the manuscript:

  1. Section 1.1: Draw a Figure summarizing TGFb signaling. This will help to the readers who are not familiarized with this pathway.
  2. Line 84: define what is MCF10A-Ca1h.
  3. Tables 1 and 2: the links shown in the table legends lead to a page with no results. Please verify them. Also, improve the format of the Tables.
  4. Table 2: it is mentioned that Results of completed clinical trials are shown in Table 2. Are there results available for these clinical trials? If so, please mention them in the table.
  5. Section 1.4: Add a new Table summarizing the clinical results described in this section.
  6. Section 1.5: Add a new Table summarizing preclinical results with inhibition of TGFb.
  7. Due to the relevance of immunotherapy studies in cancer in the last years, it would be worthy to mention if there are immunotherapy studies targeting TGFb in TNBC.

Reviewer 2 Report

Sulaiman and colleagues summarize complex matters surrounding TGF-B signaling in cancer. This is a very relevant topic that would benefit from good quality reviews. I would recommend acceptance of the paper only after addressing all the comments, especially the major comments.

Major comments

  • Please make a figure for section “1.1 Overview of TGF-B Signaling”. This section needs a figure to accompany the tekst.
  • Please tabulate the studies and their findings that describe the biphasic role of TGF-B Signaling during tumorigenesis.
  • Figure 1 is not convincing and is oversimplifying the effect of TGF-B Signaling on patient survival. A pan cancer analysis doesn’t seem appropriate, since the contribution of TGF- B Signaling is unlikely be the same across every cancer type and cancer subtype.
    • See for example: Absence of transforming growth factor-β type II receptor is associated with poorer prognosis in HER2-negative breast tumours C.E. Paiva † S.A. Drigo † F.E. Rosa F.A. Soares M.A.C. Domingues S.R. Rogatto
    • Figure 1 is currently not informative. Please show how TGF-B Signaling correlates with overall patient survival per major breast cancer subtype.
  • Line 160: “Using cBioportal and the TCGA PanCancer Atlas in our own analysis, we assessed 1082 breast cancer patients and grouped them into two categories based on TGF-β pathway gene expression (TGF-β high vs. low) [38-41]”  It’s not clear how you assessed TGF-β pathway gene expression.
  • Line 258-264: This is an example of too long sentences. Please reread the manuscript and break up sentences into shorter ones. Also use commas to help the reader.

Minor comments

  • Line 28: Introduce abbreviations. Most readers won’t know what GLOBOCAN stands for.
  • Line 33: Data is plural, so it’s “these data suggest”
  • Line 55: use “and” instead of “/”
  • Line 95: “one of the most frequently mutations in cancer”  frequently occurring
  • Line 138: “The Clinical Correlation of Dysregulated TGF-β Signaling.”  I understand what you are trying to say, however this subheading is not technically correct. Perhaps use: “Clinical correlates of…”
  • Line 267: Please reread the manuscript for sentences that appear to make sentence, however upon closer inspection do not. For example “the surviving cells possessed increased proliferation” in line 267. Cells do not possess proliferation. Similar examples are present throughout the paper.
  • Line 295: “Together this study identifies”. Together with what?
  • Line 398: “In regards to TNBC treatment, there currently exists no specific therapy. ” What does this mean?

Round 2

Reviewer 2 Report

The authors have put significant effort in revising the manuscript. I thank the authors for addressing the comments seriously.

Unfortunately still 1 comment was not fully addressed and I would strongly encourage the authors to find a way to address it. I understand it might not be easy to separate the effect of TGFb signaling on survival in the different breast cancer subtypes (especially without prior bioinformatics experience), however it is definitely possible. Learning the skills to perform this focused analysis would also help the author down the line in other projects. If the authors are not willing to do this, then they should seek a collaborator who could help them.  Once the authors can address this comment I would support acceptance for publication. 
